# Californians Linking Action with Science for Prevention of Breast Cancer (CLASP-BC)—Phase 2

**DOI:** 10.3390/ijerph17238863

**Published:** 2020-11-28

**Authors:** Jon F. Kerner, Marion H.E. Kavanaugh-Lynch, Christopher Politis, Lourdes Baezconde-Garbanati, Aviva Prager, Ross C. Brownson

**Affiliations:** 1California Breast Cancer Research Program, Bethesda, MD 20186, USA; 2California Breast Cancer Research Program University of California, Office of the President, Oakland, CA 94612, USA; Marion.Kavanaugh-Lynch@ucop.edu; 3Cancer Screening, Canadian Partnership Against Cancer, Toronto, ON M5H 1J8, Canada; Christopher.Politis@partnershipagainstcancer.ca; 4Preventive Medicine, Community Initiatives, Keck School of Medicine (KSOM), University of California, Los Angeles, CA 90033, USA; baezcond@usc.edu; 5Community Engagement, Norris Comprehensive Cancer Center, University of California, Los Angeles, CA 90033, USA; 6Center for Health Equity in the Americas, KSOM, University of Southern California, Los Angeles, CA 90007, USA; 7California Pan-Ethnic Health Network, Oakland, CA 94612, USA; aprager@cpehn.org; 8Prevention Research Center, Brown School, Washington University in St. Louis, St. Louis, MO 63130, USA; rbrownson@wustl.edu; 9Department of Surgery (Division of Public Health Sciences) and Alvin J. Siteman Cancer Center, School of Medicine, Washington University, St. Louis, MO 63110, USA

**Keywords:** implementation and dissemination, primary prevention, community-based participatory research, breast cancer, population science, action research

## Abstract

Californians Linking Action with Science for Prevention of Breast Cancer (CLASP-BC) is part of California Breast Cancer Research Program’s (CBCRP) Initiative strategic priority to disseminate and implement high-impact, population-based primary prevention interventions. CLASP-BC is informed by six years of funded program dissemination and implementation (D&I) research and evaluation conducted by the Canadian Partnership Against Cancer (CPAC) through its Coalitions Linking Action and Science for Prevention (CLASP). In its second phase, CLASP-BC will fund multi-sector, multi-jurisdictional initiatives that integrate the lessons learned from science with the lessons learned from practice and policy to reduce the risk of developing breast cancer and develop viable and sustainable infrastructure models for primary prevention breast cancer programs and research evidence implementation. Applications will be solicited from research, practice, policy, and community teams to address one or more of the intervention goals for the 23 risk factors identified in Paths to Prevention: The California Breast Cancer Primary Prevention Plan (P2P), expanding upon existing primary prevention efforts into two or more California jurisdictions, focused on disadvantaged, high risk communities with unmet social needs. The lessons learned from CLASP-BC will be widely disseminated within the participating jurisdictions, across California and, where applicable, to jurisdictions outside the state.

## 1. Introduction

Californians Linking Action with Science for Prevention of Breast Cancer (CLASP-BC) is part of the California Breast Cancer Research Program’s (CBCRP) Initiative strategic priority to disseminate and implement high-impact, population-based primary prevention interventions. By funding large-scale, evidence-informed interventions (EII), through multi-jurisdictional actions, the intent of CLASP-BC is to decrease the risk of breast cancer and other chronic diseases (sharing common risk factors), particularly among racial/ethnic minorities and medically underserved populations in California.

CLASP-BC is informed by six years of funded program dissemination and implementation (D&I) research and evaluation conducted by the Canadian Partnership Against Cancer (CPAC) through its Coalitions Linking Action and Science for Prevention (CLASP) [1,2]. CPAC issued a call for proposals from coalitions of research, practice, policy, and community-based organization partners, representing two or more provincial/territorial jurisdictions and requiring either in-kind and/or financial contributions by the grant recipient.

Each CLASP was funded for three years to implement and rigorously evaluate their evidence-informed interventions (EII) to improve the health of Canadian communities by integrating cancer prevention with strategies to prevent other chronic diseases that share common risk factors. The goal behind the CLASP initiative—and the form of the majority of the EIIs implemented through the CLASP projects—was practice and policy change resulting in a sustained impact on health beyond the CLASP funding period. A dozen CLASPs were supported over two funding rounds, including partners from every Canadian province and territory, and CPAC funds were complemented with additional funding from two partner organizations (i.e., the Public Health Agency of Canada and the Heart and Stroke Foundation of Canada).

Similarly, CLASP-BC is designed to support the dissemination, implementation, and evaluation of evidence-informed intervention (EII) strategies from the Paths to Prevention: The California Breast Cancer Primary Prevention Plan (P2P) [3] by adding value to existing cancer and chronic disease prevention efforts, focusing on identified risk factors for breast cancer. CLASP-BC will also be implemented in two phases. Phase 1 is described in detail in our first CLASP-BC publication [4].

Following the implementation and evaluation of Phase 1, Phase 2 will focus on: (1) Providing dissemination and implementation research grant support for interested and eligible coalitions demonstrating in their funding applications collaborative, evidence-informed (both practice and science-based) breast cancer prevention approaches from the P2P across two or more California jurisdictions (e.g., cities, counties); (2) working through a cooperative agreement funding mechanism, successful applicants will share the knowledge gained on quarterly calls and at annual in-person or virtual meetings (depending on COVID-19 status) to exchange ideas for how to meet the challenges and take advantage of the opportunities to sustain the breast cancer prevention approaches beyond the funding period; (3) integrating the lessons learned from science with the lessons learned from practice and policy to reduce the risk of developing breast cancer; and (4) developing a viable and sustainable infrastructure model for breast cancer prevention programs and research evidence implementation.

## 2. Background

### 2.1. CLASP-BC Coalition Formation and Evaluation

Coalitions applying for Phase 2 CLASP-BC funding will need to show evidence of inclusion (e.g., letters of commitment, letters of support, agreements to participate, sign-up sheets at meetings) with diverse community representation, and provide specific examples of how plans were developed, prioritized and selected with community engagement and buy-in. Coalitions will need to show evidence of broad community engagement and participation, including local community leaders, opinion leaders, business leaders, patient advocates, patients and their families, the public health community, local health departments, non-governmental community organizations, universities, social service agencies, and other non-profits and overall representation from priority populations that will work together for breast cancer primary prevention, public health information dissemination, research engagement, and promotion of primary prevention efforts.

The use of the principles of community-based participatory research [5] (CBPR) will be central to successful applications to the proposed CLASP-BC initiative. These include: (1) acknowledging community as a unit of identity; (2) building on strengths and resources within the community; (3) facilitating a collaborative, equitable partnership in all phases of research, involving an empowering and power-sharing process that attend to social and structural inequalities; (4) fostering co-learning and capacity building among all partners; (5) integrating and achieving a balance between knowledge generation and intervention for the mutual benefit of all partners; (6) focusing on the local relevance of public health problems and on ecological perspectives that attend to the multiple determinants of health; (7) involving systems development using a cyclical and iterative process; (8) disseminating the results to all partners and involving them in the wider dissemination of results; and (9) involving a long-term process and commitment to sustainability.

In addition to the use of the aforementioned CPBR principles in reviewing applications, elements of the conceptual framework for understanding and assessing the effectiveness of the CBPR partnership process [6] will also be adapted as review criteria for Phase 2 applications to this initiative. These elements may include environmental characteristics of the communities; structural characteristics of the coalitions; group dynamics characteristics of effective partnerships; proposed partnership EIIs; intermediate measures of partnership effectiveness; and proposed output measures of partnership effectiveness. Applicants will also be encouraged to provide evidence of how and which coalition members have worked together, and for how long, in past efforts. The evidence can be provided via letters of support or commitment, demonstrating efforts on which coalition members have worked together and the timeframe they have worked together. Not all coalition members need to have had a history of working together, but key partners with major responsibilities in the process should show some evidence of prior work in research, practice, community outreach and engagement, and/or policy implementation or advocacy. This can be demonstrated, not just related to breast cancer, but also may be demonstrated in other areas of cancer prevention, and/or other health concerns at a population level.

Both patient advocates and community organization members should be actively involved as partners to inform, promote the implementation, and evaluate the impact of policies that contribute to the reduction of breast cancer incidence in their communities. Resources from the grant can be used to support their participation in informing the dissemination and implementation of these policies, as well as their evaluation. A broad purposeful coalition will include members from organizations that may contribute in different ways to resolving some of the challenges to reduce the risk of breast cancer in the communities of focus in the Phase 2 application.

### 2.2. California Communities of Focus

CLASP-BC is designed to disseminate and implement high-impact, population-based primary prevention approaches by funding large scale EIIs, through multi-jurisdictional actions, with the intent to decrease the risk of breast cancer and other chronic diseases (sharing common risk factors), particularly among racial/ethnic minorities and medically underserved (e.g., LGBTQAI) populations in California [4].

California is exceptionally diverse. Since 2000, the state has been majority people of color, and is currently home to the country’s largest Asian and Pacific Islander community, and fifth largest African American community [7]. Nearly half of California’s population speaks a language other than English [8]. Nearly half of the state’s youth are Latinx. Estimates predict that communities of color will become an estimated 62% of the state’s population by 2030. This includes senior communities growing to an estimated 19% of California’s population in less than a decade, and including greater proportions of Latinx, Asian American, and African American. By 2045, the Latinx community is projected to make up almost half of the state’s population [7]. Therefore, the policy and planning decisions made today must have these communities’ needs in mind [9].

California’s great diversity also gives rise to great inequities. For example, the state ranks fifth in income inequality among US states [10]. California is majority people of color, yet racism and racial inequities persist [11]. These inequities manifest as higher unemployment rates for Blacks and Latinx [12]; less access to clean drinking water [13], especially for Latinx in unincorporated areas [14]; and a disproportionate impact on people of color from exposure to industrial pollution [15]. Hardship does not only affect people of color, approximately 13.5% of white Californians live in poverty and lack access to the resources that support health and well-being [7]. Many marginalized groups, including lesbian, gay, bisexual, transgender, queer, asexual, and intersex (LGBTQAI) people, and incarcerated women face their own unique challenges. These inequities are often present for people who have experienced multi-generational trauma. All these factors combined contribute to how individuals make decisions about their health and may also interfere with people’s ability to make choices about their health, due to economic, social, and structural barriers.

### 2.3. California Community Health Coalitions

There are a number of population specific coalitions focused on health in California. For example, the Latino Coalition for a Healthy California [16] has organized the state into nine regions with between one county (e.g., San Diego Regional Network) and 16 counties (Far North Regional Network) (see Figure 1) in each region.

The Latino Coalition for a Healthy California is one of four population-specific health coalitions that make up the California Pan-Ethnic Health Network [17]. The other three members are (1) the Asian and Pacific Islander American Health Forum [18], (2) the California Black Health Network [19], and the California Rural Indian Health Board [20]. There exist a number of public health efforts relevant to this project. None, to our knowledge, have focused specifically on breast cancer primary prevention.

California Accountable Communities for Health Initiative (CACHI) [21] is a vehicle for collaboration across multiple sectors to address critical community health issues. It redefines the local health system to extend beyond traditional institutions, like hospitals and health plans, and brings together clinical providers with public health departments, schools, social service agencies, community organizations, and others, in a collective effort to make a community healthier. Thirteen communities across the state are actively engaged in efforts to make changes in areas ranging from health equity, to asthma, cardiovascular disease and substance use.

The California Endowment [22] has for a decade supported 14 localities around California in a broad array of systems and policy change targets to reshape the places that shape people’s health—their neighborhoods. While focused in specific communities, the collective learning and energy from these communities contribute to statewide policy and systems change to promote health, health equity, and health justice. In other words, it is a place-based strategy, but with an attitude for statewide change. One of the major tenets has been a focus on prevention, although cancer prevention, and more specifically, breast cancer prevention, has not been a target.

In February 2019, the San Joaquin Valley city of Stockton, which bridges the Bay Area and the Central Valley, began providing 125 residents with monthly payments of $500 as a donor-funded pilot of guaranteed basic income [23]. The success of the pilot has led Stockton’s mayor to start Mayors for a Guaranteed Income. Evaluation has revealed direct connections to health, as it has been found that participants were using their additional cash to invest in healthier food choices for themselves and their children, evidence that the barrier for lower-income Americans to eat healthier is not just education about what they should and should not be eating, but access to affordable, high-quality food [24].

The California Tobacco Partnerships [25,26] (also known as statewide ethnic networks), worked with communities to support tobacco prevention, cessation and control at the grass roots level in the African American, Hispanic/Latinx, American Indian, and Asian/Pacific Islander communities. In later years new groups were added including the LGBTQAI and low socioeconomic status non-Hispanic White populations, and a youth coalition, the California Youth Advocacy Network funded by the State of California. With declining funding, priority population training and technical assistance services were later consolidated into a single statewide project with a flexible organizational structure intended to be responsive to cross-cutting (e.g., the culture of poverty, low literacy) and emerging population-specific needs (e.g., mental illness). Since 2006, a variety of other tobacco control networks for tobacco control and prevention were patterned after these, and funded at a national level by the Centers for Disease control and several still exist [27]

In 2011, the ADEPT Coalition [28] was formed and has continued to exist with non-State funding. ADEPT is the Advocacy and Data Dissemination to Achieve Equity for Priority Populations on Tobacco (ADEPT), a California multiethnic and cross-cultural equity collaborative focused on tobacco control. Members include organizations and individuals from the Asian/Pacific Islander Community, Hispanic/Latinx, African American, American Indian, and LGBTQ communities. ADEPT promotes integration of health equity principles in decision-making bodies, building partnerships with equity collaboratives, fostering collaborations across health justice movements, and supporting policy goals. Thus, as one of the most populous and most diverse states in the country, California has a rich history of health coalition building, community leadership and advocacy upon which CLASP-BC can build.

### 2.4. Lessons Learned from the Canadian CLASP Experience

CPAC funded 12 pan-Canadian Coalitions Linking Action and Science for Prevention (CLASP) from 2009 to 2014. CPAC’s pan-Canadian CLASP framework focused on engaging communities from 10 Canadian provinces and three territories with an emphasis on First Nations, Inuit, and Métis communities. Canada’s Indigenous population experiences a disproportionate cancer burden compared to the rest of the Canadian population due to a history of colonial policies that have had negative long-term effects on First Nations, Inuit, and Métis physical and emotional well-being [29]. To have a significant and equitable impact on cancer prevention in Canada, it was important to prioritize proposals and partnerships that would focus on Indigenous-specific cancer prevention. This was accomplished through the Call for Proposals scope and proposal adjudication review criteria and resulted in seven of the 12 CLASP projects that included cancer prevention strategies developed with and for First Nations, Inuit, and Métis communities.

As such, the lessons learned from CPAC’s CLASP investments can help inform the design of the proposed CLASP-BC initiative. However, the new and innovative elements of this initiative will lead to California being the first state in the U.S. making such a substantial commitment to, and investment in, integrating the lessons learned from science with the lessons learned from practice and policy to reduce the risk of developing breast cancer.

In Canada, CLASP was developed to respond to the broad cancer prevention needs across the country and, therefore, a wide array of modifiable risk factors was addressed through the initiative with a majority of the projects focused on reducing risk in children and youth. Two CLASPs specifically focused on policy approaches addressing breast cancer risk factors in adults.

Healthy Canada By Design (focused on promoting increased physical activity and reducing environmental pollution) and POWER-Up! (focused on postnatal breastfeeding and nutrition policy approaches to reduce obesity) combined to integrate cancer prevention within 164 new and revised policies at the provincial/territorial, municipal, and school level through multidisciplinary collaboration [30,31]. The lessons learned from Healthy Canada By Design and POWER-Up are described below. Additional examples of policy approaches to cancer prevention and public health can be found in CPAC’s Prevention Policy Directory—an online tool developed to support cross-jurisdictional policy diffusion and was utilized to evaluate policy impact of the CLASP initiative [32].

Healthy Canada by Design and POWER-Up! both influenced municipal and provincial/territorial policies through two key mechanisms: facilitating cross-disciplinary collaboration and developing of policy tools.

#### 2.4.1. Facilitating Cross-Disciplinary Collaboration

In the case of Healthy Canada by Design, public health, land use planning, and transportation engineering experts were brought together within municipal and provincial/territorial governments to share technical knowledge and experience and ultimately integrate a health and cancer prevention lens within land use and transportation planning policies. This was done through formal and sustainable mechanisms that broke down existing silos among different disciplines, such as planning staff seconded to public health units, public health staff sitting on land use advisory committees, and changes to planning policy review cycles to include Medical Officers of Health [33]. In an analysis of 163 built environment policies to promote physical activity developed through CLASP projects 14, ‘critical success factors’ were identified. Eight of the 14 ‘critical success factors’ were dependent on multidisciplinary collaboration illustrating the catalyzing role CLASP coalitions can play in cancer prevention policy development [34]. Examples of policy outcomes from Healthy Canada by Design’s cross-pollination of skills, expertise, and collaborative approach to working included:

An amendment to the Regional Municipality of Peel’s (which includes two of the largest cities in Canada: Mississauga and Brampton) Official Plan [35] to include:Health impact assessments as a requirement for development applications;Public health indicators to analyze the effectiveness of Official Plan policies and serve as a basis for policy adjustments;All development applications to have regard for public health.

City of Fredericton’s City Centre Plan Update [36] and Main Street Guidelines [37] were developed in collaboration with the provincial Medical Officer of Health to include entire sections and priorities focused on healthy communities and healthy built environment.

In addition to these dissemination and implementation strategies, Healthy Canada By Design also supported smaller “hybrid” evaluation studies where tools and resources developed primarily for larger urban contexts were adapted and re-evaluated in rural and remote communities with more limited resources to address cancer and chronic disease prevention priorities.

#### 2.4.2. Development of Policy Tools

Both Healthy Canada by Design and POWER-Up! were able to influence the creation of healthy public policies for physical activity and food environments by developing policy tools that were evidence-informed yet designed for easy and seamless uptake by policymakers. These tools, such as health impact surveys and model policies, were built to integrate a health promotion and cancer prevention lens directly into municipal planning practice and policymaking; but were also developed to meet broader needs to ensure uptake and sustainable use beyond project funding. Examples of policy tools include:

The Health Background Study Framework was created through Healthy Canada by Design to support consideration of health impacts within the Regional Municipality of Peel and the City of Toronto’s approval process for land use development [38]. The Framework was created to be simple and instructive with applicability to a range of development locations, scales, and stages of the development process. The Framework ensures health considerations are made in land use development whether health experts are engaged or not.

A software-based health impact assessment tool [39], created through Healthy Canada by Design, links detailed measures of walkability and regional accessibility with travel, physical activity, health indicators and greenhouse gas (GHG) emissions for land use developments in the City of Toronto [40]. The tool is used to model health and environmental impacts of proposed land developments and was applied to select an optimal design for the 2015 Pan Am Games Athletes’ Village (now known as the West Don Lands neighborhood) in Toronto. Using the tool, land use planners and transportation engineers were able to design a community where walking and biking trips would be more than double, and transit trips increased by one third, in comparison to typical developments. The tool also predicted per capita automobile trips would be cut in half, and vehicle kilometers travelled and GHG emissions would decrease by 15% and 29%, respectively.

Model municipal policy resolutions were developed through POWER Up! to create healthier food environments by enabling breastfeeding in public places, restricting access to sugar-sweetened energy drinks, and increasing access to drinking water. The Coalition Poids (i.e., Weight Coalition) in Quebec worked with municipal leaders to understand policy gaps and needs and then engaged legal expertise and public health researchers to draft policies that could be enacted across the province—effectively centralizing municipal policymaking within their organization to reduce duplication of efforts. Policy outcomes from this approach include:
-Energy Drink Sales Ban in Public Buildings’ [41] policy to restrict the sale of sugar-sweetened energy drink beverages in public buildings and promote healthier alternatives was adopted in 83 municipalities.-Access to Drinking Water’ [42] resolution to set standards around access to free and clean drinking water in public spaces and create healthier alternatives to sugar-sweetened beverages was adopted in seven municipalities.-Making Municipalities More Breastfeeding Friendly Resolution [43] to make environments hospitable and friendly to breastfeeding mothers and facilitate the practice of breastfeeding anywhere and anytime was adopted in at least one municipality.


## 3. Future Directions for Research

### 3.1. Phase 2: CLASP-BC Dissemination and Implementation Research Projects

In Phase 2 of CLASP-BC we intend to solicit applications from research, practice, policy, and community teams to:-Address one or more of the intervention goals for the 23 risk factors identified in P2P [3];-Expand upon existing primary prevention efforts into two or more California jurisdictions;-Focus on disadvantaged, high risk communities with unmet social needs;-Actively engage the leadership of local community-based organizations with research scientists, public health and/or community health practitioners, and legislative/executive policy influencers/makers as partners; and rigorously evaluate the impact of these expanded collaborative efforts;-Include and update annually a sustainability plan for successful dissemination and implementation approaches;-Collaboratively disseminate results of D&I research through community, practice, and policy presentations and policy briefs (e.g., social media, press conferences, town hall/community meetings, press release, policy briefs, newsletters and magazines), as well as peer-reviewed publications.

Applicants will be encouraged to leverage existing initiatives in California (e.g., the Building Healthy Communities initiative sponsored by the California Endowment) [44], as well as federal-, state-, and county-funded research organizations (e.g., NCI-designated Comprehensive Cancer Centers). Coalitions that provide synergy and evidence for inclusion tend to be more successful in the planning of dissemination, implementation, and evaluation as there is greater buy-in from different constituents. For there to be robust community engagement and participation, materials and planning should be as accessible as possible. This includes, but is not limited to, materials that are culturally and linguistically appropriate, ways to participate that account for varying access to technology (e.g., providing Zoom information and a call-in only options), and a reliance on trusted community members to lead outreach and education efforts.

A description of joint collaborative and sustainable partnerships and processes will be needed in order to support the activities of the grant. Specifications of partners and what each brings to the table will be preferred, as well as avoidance of duplication of efforts. Ways in which each partner contributes will need to be clearly outlined. The potential synergistic effect that will be generated by the proposed combination of agency partners should be explained.

To ensure that every funded proposal is poised for a successful partnership, the review panel(s) must be as multi-sectoral, multi-jurisdictional and collaborative as the proposed coalitions and the evaluation criteria and scoring developed to measure these elements. This can be modeled on the very successful Community Research Collaboration Program of CBCRP, which was itself based largely on Larry Green’s *Study of Participatory Research in Health Promotion: Review and recommendations for the development of participatory research in health promotion in Canada* [45]. While his recommendations were not taken up by any other funding agency to our knowledge, they have provided an excellent framework to CBCRP in selecting just those applications that marry scientific rigor with community involvement and benefit. In the 25 years CBCRP has funded community-partnered participatory research, the scoring criteria applied by a very diverse committee of academic and community researchers has identified research that is simultaneously scholarly, community-driven, and collaborative.

### 3.2. Lessons Learned from the CPAC Funding Management Experience in Canada

During the funded Phase 2 CLASP experience in Canada, several additional challenges were encountered consistently across most of the coalitions.

#### 3.2.1. Research versus Knowledge to Action

In some cases, the focus on knowledge to action and implementation, in comparison to the more traditional research grant, was a challenge for research-based partners. The balance between adherence to an ideal research design and contextualization for real-life uptake in a program or policy is difficult to strike and requires creativity and negotiation among research, practice, policy, and community partners. This was especially important in that adaptation [46] to local contexts was a key driver of implementation success [29].

#### 3.2.2. Capacity for Reporting and Evaluation

A key learning from the Canadian CLASP initiative was the importance of considering equity in capacity across the partner disciplines. Community-based practice organizations and indigenous communities often lacked the capacity to balance in-depth quarterly reporting and ongoing evaluation requirements in addition to project implementation [47].

#### 3.2.3. Coalition Building and Policy Change Takes Time

CLASP projects were working within three-year funding periods—an ambitious timeline to effect policy change within any level of government. Factoring in that many coalitions also need to build partnerships and trust among partners, policy impact at the end of a three-year project becomes less likely. Despite the significant policy impacts (discussed earlier) reported by CLASP projects at the end of their projects, anecdotal reports of continued policy success were received for years after the CLASP initiative concluded [32].

Some strategies were devised through the CLASP initiative in Canada to address these issues. Other strategies were borne out of CLASP follow-up evaluation efforts and applied to later Canadian cancer prevention initiatives. Key strategies that may be considered for CLASP-BC include:Small-scale planning funds to support community and indigenous organizations with a project manager to draft a proposal: A key barrier to community-based partner participation identified in post-CLASP evaluation efforts was a lack of resources available to develop proposals and/or a lack of resources knowledgeable in proposal writing. This has since been addressed in some new initiatives by providing potential partners with small-scale funds (approximately 25,000 $CDN) to contract an experienced project manager to develop a proposal. This project manager can often be retained for the project itself to support implementation. The need for this support is typically identified through an earlier letter of intent (LOI) that provides high-level details of the project’s approach and how planning funds will be used to further develop the proposal. A key challenge with this approach is whether there are experienced project managers available—in some rural and remote areas of Canada recruitment was difficult even when funds are available. Given broader on-line access and working from home restrictions during the COVID-19 pandemic, in California these geographic challenges may not be as much of an issue. With respect to CLASP-BC, the pre-application training in dissemination and implementation research, and CBPR, during Phase 1 [4] may help address some of these concerns.Additional feedback/review cycles from the funding organization staff to provide feedback and guidance on early drafts of a proposal: In this process, more time was allocated to the Phase 2 proposal development phase with several iterations of the proposal going back and forth between funder and partners to build on each other’s ideas. The result was a proposal that the funder was confident met the initiative objectives and a greater understanding of the desired outcomes by the partners, while not guaranteeing any particular outcome from the outside peer review process. This administrative review process was accomplished by a program manager at the funding organization working with the partners via email and teleconference. This supported lower capacity partners in proposal development. Through these email and teleconference interactions, it was also an opportunity to ensure that the focus on knowledge to action was clear and, if indigenous partners were to be engaged, there was sufficient expertise in the coalition to do this in a meaningful way.Research/practice/policy/community peer reviewer orientation: One challenge to transdisciplinary collaboration is that research, practice, and policy representatives often have disparate perspectives on what is required to solve complex health problems [48,49]. To reduce systematic variability in ratings because of discipline-specific perspectives, the Partnership hosted a two-day orientation meeting prior to the proposals being distributed to individual members of the adjudication panel [2]. The orientation familiarized the adjudicators with the review procedures and assisted them in finding common ground about how to apply the review criteria. During the two-day orientation, adjudication panel members were grouped and regrouped into discussion tables that included research, practice, and policy members based largely on the proposals to which they were to be assigned. Group discussions specifically focused on identifying where research, practice, or policy perspectives may differ as applied to each of the review criteria, and how those discipline-specific differences might be bridged in the peer review process.Ongoing project support: In post-CLASP evaluation feedback, partners were very supportive of the hands-on support they received during project implementation. Monthly check-in calls were held between the funder and coalition leads that provided an opportunity to discuss project progress, but also to co-problem solve potential issues and identify new opportunities.Evaluation: Evaluating the impact of any cancer prevention initiative is critical. Data and findings around practice and policy outcomes from the CLASP projects were submitted at the conclusion of the funding period, but many policy impacts occurred after the funding period had concluded given the time-scale involved in building multidisciplinary coalitions and influencing policy processes. This is captured, in part, in publications by CLASP partners post-funding. A follow-up evaluation period two to three years after the conclusion of implementation funding could be an opportunity to capture the sustained results of the funded policy work.

## 4. Application to Breast Cancer Prevention

### 4.1. Opportunities for Breast Cancer Primary Prevention Interventions

Since 1998, the Centers for Disease Control and Prevention (CDC) has supported all 50 states, the District of Columbia, six U.S. Associated Pacific Islands and Puerto Rico, and eight tribes or tribal organizations create and implement cancer control plans [50]. In regard to breast cancer, these state plans emphasize early detection, treatment, and access to services. Where they do address primary prevention, the plans tend to be focused on recommended ways for individuals to change their behavior without significant consideration of social, environmental, and/or situational factors or obstacles that enhance or limit individual efforts. This is true for California, and has been true for the focus of most state-level and community-level breast cancer efforts.

Paths to Prevention: The California Breast Cancer Primary Prevention Plan (P2P), whose development has been described elsewhere [3,4], provides an action plan of local, regional, and statewide evidence-informed interventions, both research-based and practice-based, to foster an environment that reduces a community’s risk for breast cancer. The plan is unique from other cancer plans in several important ways, including its sole focus on primary prevention, its focus on systemic interventions, the social justice lens through which breast cancer is viewed and the weaving together of scientific and community wisdom. P2P is based on the realization that breast cancer risk is not simply about the individual; it is about a society that has the potential to function in a way that promotes people’s health.

P2P provides a summary of the scientific knowledge and intervention goals and strategies to affect change on 23 risk and protective factors, including place-based chemicals; ionizing and non-ionizing radiation and other physical factors; physical activity; alcohol consumption and tobacco use; as well as systemic oppression and poverty. While the focus of the P2P is squarely on breast cancer, the policy action plan will undoubtedly impact numerous other cancers and adverse health impacts. Moreover, while the focus is squarely on California, the plan will provide a road map for other states to prioritize primary prevention of breast cancer.

The goals and interventions within the Paths to Prevention: The California Breast Cancer Primary Prevention Plan (P2P) can be complemented by a variety of guidelines and systematic reviews of community-level interventions. Among the most relevant guidelines for prevention of breast cancer are the *Guide to Community Preventive Services* (the Community Guide) [51], the National Cancer Institute’s *Evidence-based Cancer Control Programs* (formerly RTIPs) [52], and *What Works for Health*, which is supported by the Robert Wood Johnson Foundation [53]. These guidelines commonly provide information on effective programs and policies for practitioners and other stakeholder. Every guideline is different and has strengths and limitations. The Community Guide follows a rigorous and time-intensive process. In RTIPs, programs are rated across the RE-AIM framework to assess the potential for implementation and long-term impact. *What Works for Health* relies not only on peer reviewed literature but also on expert opinion when making intervention recommendations. Roughly half of local-level public health practitioners report use of the Community Guide in their work, suggesting substantial room for improvement [54].

### 4.2. Phase 2 Dissemination Plans to Enhance Breast Cancer Prevention

#### 4.2.1. Methods to Ensure Application of Findings

Each CLASP-BC Phase 2 coalition applicant will be expected to present their plans to disseminate the results to all coalition partners and involve them in the wider dissemination of results to project funders, as well as local and state stakeholders and policy decision makers. In addition, new and evolving models (e.g., social media) that enhance dissemination [55] should be described in a competitive Phase 2 application. The funder may, in addition, choose to fund additional dissemination of the most successful projects. This is currently done through CBCRP’s Policy Initiative, in which short, rapid-response policy research projects are often supplemented after completion by a targeted dissemination effort that is crafted by the research team with the advice of the Policy Research Advisory Group.

#### 4.2.2. Potential Impact on Policy

While policy dissemination research is relatively under developed in the field of health, policy dissemination research in other areas is not a new field and is more developed in countries outside the United States [56]. The lessons learned from this research, as appropriate, should be included in a Phase 2 application.

#### 4.2.3. Translational Potential

To the extent appropriate, successful applications will describe how the lessons learned from the specific CLASP-BC project, in specific California jurisdictions, will be translated across the state and may be replicable in other jurisdictions outside the state of California.

### 4.3. Evaluation

It is the practice of CBCRP to develop an evaluation plan based on a logic model that addresses the short-, mid- and long-term goals of any new funding opportunity. Subsequent evaluations are conducted by internal and/or external evaluators and published in peer-reviewed or self-published publications [57].

## 5. Conclusions

This concept paper describes the second phase of a planned research funding initiative by the California Breast Cancer Prevention Program entitled Californians Linking Action with Science for Prevention of Breast Cancer (CLASP-BC). CLASP-BC is designed to translate the overarching Paths to Prevention: The California Breast Cancer Primary Prevention Plan (P2P) goal and specific intervention goals for 23 breast cancer risk and protective factors strategies into evidence-informed interventions that are disseminated and implemented across California. CLASP-BC is designed to disseminate and implement high-impact, population-based primary prevention approaches by funding large scale evidence-informed interventions, through multi-jurisdictional actions, with the intent to decrease the risk of breast cancer and other chronic diseases (sharing common risk factors), particularly among racial/ethnic minorities and medically underserved populations in California.

CLASP-BC will be rolled out in two phases. Phase 1 has been previously described [4]. Phase 2 will focus on: Addressing one or more of the intervention goals for the 23 risk factors identified in P2P [3]; expanding upon existing primary prevention efforts into two or more California jurisdictions; on disadvantaged, high risk communities with unmet social needs; actively engaging the leadership of local community-based organizations with research scientists, public health and/or community health practitioners, and legislative/executive policy influencers/makers as partners; and rigorously evaluating the impact of these expanded collaborative efforts; including and updating annually a sustainability plan for successful dissemination and implementation approaches; and collaboratively disseminating the results of this D&I research through community, practice, and policy presentations, and policy briefs (e.g., social media, press conferences, town hall/community meetings, press release, policy briefs, newsletters and magazines), as well as peer-reviewed publication.

This paper describes in detail the plans for implementing CLASP-BC Phase 2. The Phase 2 elements of CLASP-BC are predicated on the successful implementation and evaluation of Phase 1 [4].

## Figures and Tables

**Figure 1 ijerph-17-08863-f001:**
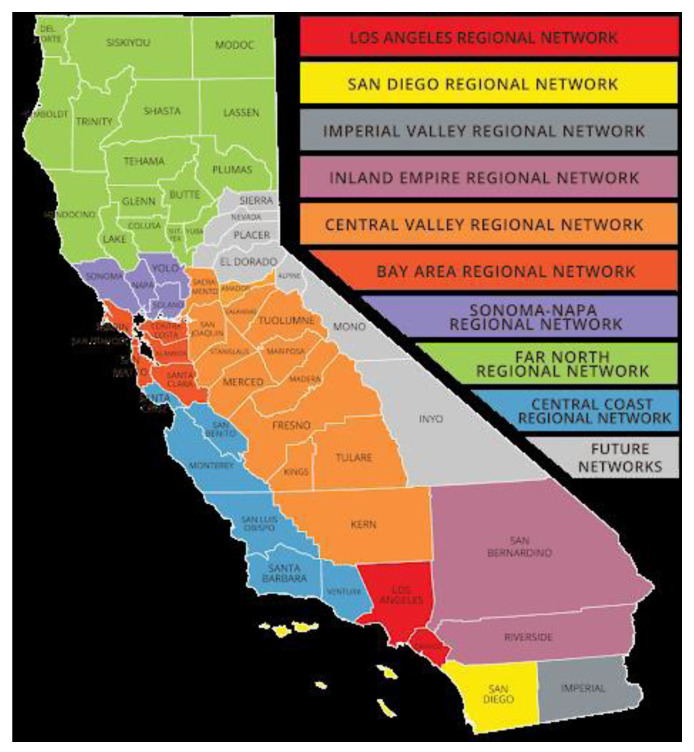
Regional Network for Latino Coalition for A Healthy California.

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
