# Peer review of "Californians Linking Action with Science for Prevention of Breast Cancer (CLASP-BC)—Phase 2"

_ijerph, 2020, doi:10.3390/ijerph17238863_

Round 1

Reviewer 1 Report

Kerner JF et al. described a concept paper of Californian Linking Action with Science for Prevention of Breast Cancer (CLASP-BC) phase 2.  The background of CLASP-BC and contents are described clearly and in detail.  Because the manuscript is not an original article but a concept paper, the contents are sufficient to inform the initiative of phase 2 to academic members.  This reviewer hopes that the outcome of the 2nd phase will be reflected in public policy and helpful for welfare of people regardless of socioeconomical statuses.

The background is too lengthy, and the part that is not directly related with this subject and has already been presented in their previous manuscript (IJERPH 2020, 17, x, doi is not provided in References) could be omitted.

Author Response

Our thanks to reviewer #1 for the review feedback on our Phase 2 CLASP-BC concept paper.  We share the hopes that the implementation of Phase 2 will lead to major policy changes to reduce the risk of developing breast cancer and other chronic diseases that share common risk factors. 

With respect to the concern that "the background is too lengthy, and the part that is not directly related with this subject and has already been presented in their previous manuscript (IJERPH 2020, 17, x, doi is not provided in References) could be omitted," we deleted paragraphs in the introduction, background and future directions for research sections of the manuscript that had been previously addressed in our Phase 1 concept paper.  We also added the doi information to reference #4. 

Reviewer 2 Report

Manuscript authors by Jon F. Kerner et al. discusses the a planned research funding initiative by the California Breast Cancer Prevention Program entitled Californians Linking Action with Science for Prevention of Breast Cancer (CLASP-BC). CLASP-BC is informed by CBPR and D&I research and is designed to translate the overarching Paths to Prevention: The California Breast Cancer Primary Prevention Plan 5(P2P) goal and specific intervention goals for 23 breast cancer risk and protective factors strategies into evidence-informed interventions that are disseminated and implemented across California..
This subject was not undertaken never on this comprehensive scale. The authors describe in detail the plans for implementing CLASP-BC Phase 2. The Phase 2 elements of CLASP-BC are predicated on the successful implementation and evaluation of Phase 1.
Due to the lack of comprehensive study on this theme, this publication is very innovative and is an indication for further research on this problem. The manuscript may be accepted for publication.

Author Response

We thank reviewer #2 for the positive review comments of our Phase 2 CLASP-BC concept paper and very much appreciate the recommendation to accept our manuscript for publication. 

Reviewer 3 Report

Dear Authors,

Question 1: Insert figure 1  on "California community health coalitions".

Att

Reviewer

Author Response

As requested by Reviewer #3, we have added "Figure 1" as a title above the figure itself on line 170 of the manuscript.  We thank Reviewer #3 for catching this omission. 

Reviewer 4 Report

This paper entitled “Californians Linking Action with Science for 2 Prevention of Breast Cancer (CLASP-BC) - Phase 2”, is very well written.  This is a very interesting paper in that it describes itself as a concept paper, which it is, that describes a planned research funding initiative by the California breast cancer prevention program that is entitled “Californians Linking Action With Science For The Prevention Of Breast Cancer”. This is a highly detailed paper focused on the description and implementation of the phase 2 of this program which will focus on 4 main aspects. First focuses on grant support those interested and for collaborative studies. The second focuses on the exchange of ideas through cooperation of the researchers. Third, nicely described, the coupling of science lessons learned with those of practice/policy in order to reduce the risk of developing breast cancer. The last, also nicely presented, as the foresight of developing a viable and sustainable infrastructure model for breast cancer prevention programs.

All, in all, this paper presents very important ideas with regards to breast cancer and of phase 2 of a very important program.

Author Response

We thank reviewer #4 for this very positive review of our manuscript. We very much appreciate the reviewer's summary that "all, in all, this paper presents very important ideas with regards to breast cancer and of phase 2 of a very important program."

Reviewer 5 Report

This a very interesting manuscript which a program from the State of California to disseminate and implement high-impact, population-based primary prevention interventions. The main problem with this manuscript is that it does not correspond to any of the 3 main types of works that typically accept IJERPH (original research manuscripts; reviews; case reports).

The structure of the text also does not follow the sections required by IJERPH (Introduction, Materials & Methods, Results, Conclusions). This is very unfortunate because the work described in this manuscript is highly relevant in public health. I strongly encourage the authors to transform this manuscript in the way is required by a "classical" scientific paper. The format authors have chosen is obviously a policy report and as such can not be published in IJERPH.

I have been once again reviewing this manuscript. It´s an interesting text which describes a complex initiative to disseminate and implement high-impact, population-based primary prevention interventions in the state of California.
Although the key-words for this paper are action-oriented (implementation and dissemination; primary prevention; community-based participatory research; breast cancer; population science; action research) and one may expect a description of a series of interventions, their results, and a discussion on their success or not, this is not the case. What this paper represents is the rationale for applications that are going to be solicited from research, practice, policy, and community teams to address one or more of the intervention goals for the 23 risk factors identified in Paths to Prevention. This manuscript ends up indicating that those projects will be Phase 2 of this initiative. Phase 1 has been already published.
The only option I see for this paper to be considered for publication is if authors decide to provide concise and precise updates on the latest progress made in the area of research in which the projects are going to be submitted and transforming the paper into a Review.

Author Response

We thank Reviewer 5 for the review of our manuscript. The reviewer suggests restructuring the format of the paper into a regular paper structure typically accepted by IJERPH (original research manuscripts; reviews; case reports) or a Review structure.  However as the section editor has noted, the current format flows well and matches the needs of this informative "Concept Paper."

The reviewer notes that "what this paper represents is the rationale for applications that are going to be solicited from research, practice, policy, and community teams to address one or more of the intervention goals for the 23 risk factors identified in Paths to Prevention. This manuscript ends up indicating that those projects will be Phase 2 of this initiative." While our Phase 1 concept paper has indeed been published, the Phase 1 funding initiative is in the process of being implemented. As such, the implementation of the Phase 2 research funding initiative described in this concept paper awaits the successful completion of the Phase 1 funding initiative.

We look forward to submitting research review manuscripts in the future to IJERPH providing "concise and precise updates on the latest progress made in the area of research in which the projects are going to be submitted" when each phase of CLASP-BC funding initiatives are completed.

Round 2

Reviewer 5 Report

The only option I saw for this paper to be considered for publication is if the authors decided to provide concise and precise updates on the latest progress made in the area of research in which the projects are going to be submitted and transforming the paper into a Review.

This paper can not describe as authors indicate the second phase of the planned research initiative as they claim, because the second has not already started. I can not accept this paper.